# The Crystal Structure Elucidation of a Tetrapeptide Analog of Somatostatin DOTA-Phe-D-Trp-Lys-Thr-OMe

**Sabina Diusenova** [1,2], **Sergey Arkhipov** [2,3,4] , **Dmitry Avdeev** [5], **Pavel Dorovatovskii** [6], **Derenik Khachatryan** [7], **Vladimir Lazarenko** [6], **Michael Medvedev** [8] , **Alena Nikolaeva** [6], **Mikhail Ovchinnikov** [5] , **Maria Sidorova** [5] and **Yan Zubavichus** [4,* ]

1 Boreskov Institute of Catalysis SB RAS, 630090 Novosibirsk, Russia; s.diusenova@g.nsu.ru
2 Faculty of Natural Sciences, Department of Solid State Chemistry, Novosibirsk State University, 630090 Novosibirsk, Russia; arksergey@gmail.com
3 State Research Center of Virology and Biotechnology VECTOR of the Federal Service for Surveillance in Consumer Rights Protection and Human Well-being (FSRI SRC VB VECTOR), 630559 Koltsovo, Russia
4 Synchrotron Radiation Facility SKIF, G.K. Boreskov Institute of Catalysis SB RAS, 630559 Koltsovo, Russia
5 National Medical Research Center for Cardiology, Ministry of Health of the Russian Federation, 121552 Moscow, Russia; peptides.dv@gmail.com (D.A.); peptide-mv@mail.ru (M.O.); mvs.peptide@gmail.com (M.S.)
6 National Research Center, Kurchatov Institute, 123189 Moscow, Russia; paulgemini@mail.ru (P.D.); vladimir.a.lazarenko@gmail.com (V.L.); nikolaeva_ay@nrcki.ru (A.N.)
7 Institute for Chemical Reagents and High Purity Chemical Substances of NRC, 107076 Moscow, Russia; hr.peptides@gmail.com
8 N. D. Zelinsky Institute of Organic Chemistry, Russian Academy of Sciences, 119991 Moscow, Russia; medvedev.m.g@gmail.com
* Correspondence: ya.v.zubavichus@srf-skif.ru; Tel.: +7-913-201-41-44

**Abstract:** Herewith, we report for the first time the crystal structure of tetrapeptide FwKT (Phe-D-Trp-Lys-Thr), which is considered to represent an epitope for biomedically relevant hormone somatostatin. The target molecule was successfully crystalized, solved and refined as a conjugate of the tetrapeptide moiety bearing a protective group DOTA at the N-terminus and methylated at the O-terminus. The combination of a hormone active site and a powerful chelator make the substance a highly prospective targeted drug delivery system, especially for peptide receptor radionuclide therapy (PRRT) applications.

**Keywords:** XRD; analogue; somatostatin; DOTA; PRRT

## 1. Introduction

This work is a continuation of our research on the search and synthesis of new analogs of the hormone somatostatin, which is secreted mainly in the pancreas and the anterior pituitary gland. Somatostatin has a wide spectrum of biological activity and inhibits many physiological functions in the hypothalamus, gastrointestinal tract and pancreas [1–3]. Currently, somatostatin analogs are used in medicine in the therapy of hormone-active tumors in order to suppress the secretory activity of tumor tissue. There are a large number of drugs on the pharmaceutical market based on various somatostatin fragments [4]. A widely used radiopharmaceutical in nuclear medicine is a somatostatin analog—a complex of octreotide conjugate and 1,4,7,10-tetraazacyclododecane-1,4,7,10-tetraacetic acid (DOTA) with $^{177}$Lu-"Lutathera"; this drug is used for radiation therapy of neuroendocrine tumors [5]. Note that most of the therapeutic analogues of somatostatin are developed on the basis of its pharmacophore sequence: -Phe-Trp-Lys-Thr-, which is responsible for their binding to somatostatin receptors (sstr), the overexpression of which is observed in tumor cells. According to the literature data, somatostatin and drugs based on it in the most stable conformation take the structure of a β-hairpin closed at the base by a disulfide bridge (S-S), which is responsible for higher selectivity to sstr and stabilization of the structure. It is the

conformation of the β-hairpin that is the fundamental factor for the stabilization of their structure and high selectivity of action to somatostatin receptors [6].

Previously, we have synthesized chimeric molecules containing in their structure the pharmacophore sequence -Phe-D-Trp-Lys-Thr-OMe, which selectively acts on human lung adenocarcinoma (A549) (IC50(Human embryonic fibroblasts (HEF))/IC50(A-549) > 9) [7]. In addition, we studied the labeling of the conjugate described in this work with $Sc^{3+}$, $Bi^{3+}$ and $Eu^{3+}$ cations and showed the stability of these complexes in solutions with biologically relevant ions and in bovine serum at 37 °C [8]. In this work, we attempted to study the conformation of the -DOTA-Phe-D-Trp-Lys-Thr-OMe conjugate to understand the structure–activity relationship using X-ray structural analysis (XRD) data and quantum chemistry methods. Note that currently few crystal structures of somatostatin hormone analogues studied by XRD have been described [9,10], as obtaining a crystal of a peptide conjugate and further studying the crystal structure of peptide molecules is a non-trivial task and an important factor in the development and targeted design of peptide inhibitors. This article is the very first structural report on a peptide linear analogue somatostatin molecule.

## 2. Experimental

### 2.1. Synthesis

The peptide Phe-*D*-Trp-Lys(ε-Boc)-Thr-OMe (I) in this work was synthesized according to the previously described method [11]. Below is a scheme for preparing a conjugate of peptide I with a DOTA chelator. When creating an amide bond in conjugate II, the carbodiimide method was used. The crude product III was obtained with a purity of over 90%, which facilitated its purification by preparative high performance liquid chromatography (HPLC) to a purity of 98.6%. Compound III was characterized by mass spectrometry and X-ray diffraction analysis (Scheme 1).

**Scheme 1.** Synthesis scheme for (**I**)—H-Phe-D-Trp-Lys(ε-Boc)-Thr-OMe; (**II**)—DOTA-Phe-D-Trp-Lys(ε-Boc)-Thr-OMe; (**III**) DOTA-Phe-D-Trp-Lys-Thr-OMe.

### 2.2. Crystallization

The functionalized FwKT tetrapeptide was crystallized by the "hanging drop" vapor diffusion method in 24-well VDX plates (Hampton research, Aliso Viejo, CA, USA). A load of 1.5 μL of the tetrapeptide sample was mixed with 1.5 μL of precipitant also containing 100 mM of HEPES (4-(2-hydroxyethyl)-1-piperazineethanesulfonic acid) buffer to maintain pH at 7.5 and 25–31% *w/v* PEG 3350. The crystallization liquor was set up over 500 μL of the precipitant in a sealed reservoir. The crystallization plates were incubated for a week at 15 °C. After that they were regularly inspected for the crystal growth by a remotely controlled digital camera. Crystals suitable for a crystallographic study grew in 8–12 days. The crystals were characterized by rod-like shapes and maximum linear size of 100–200 μm. Immediately before the diffraction data collection, the crystals were briefly soaked with the precipitant solution supplemented with 25% glycerol as a cryoprotectant and flash-frozen in liquid nitrogen.

### 2.3. X-ray Diffraction Data Acquisition and Analysis

Single-crystal diffraction datasets for DOTA-Phe-D-Trp-Lys-Thr-OMe were acquired at the BELOK beamline of the Kurchatov Synchrotron Radiation Source (NRC "Kurchatov Institute", Moscow). X-ray radiation with λ = 0.80246 Å was used; the distance between the

crystal and a Rayonix SX-165 CCD detector was set to 43 mm at normal orientation [12,13]. The temperature was kept at 100(2) K during the measurements using an Cryostream-type LT-device, (Oxford Cryosystems Ltd., Oxford, UK). Diffraction frames were collected in the φ-scanning mode with an oscillation range of 1°. The frames were indexed, integrated and scaled using the *XDS*, 5 February 2021 version; MPI for Medical Research: Heidelberg, Germany, 2021 [14]. The apparent resolution is 0.78 Å.

The structure was solved by the intrinsic phasing modification of direct methods as implemented in *SHELXT*, 2018/2 version; Georg-August Universität Göttingen: Göttingen, Germany, 2018 [15] and refined by the full-matrix least-squares refinement against $F^2$ with the *SHELXL*, 2017/1 version; Georg-August Universität Göttingen: Göttingen, Germany, 2017 [16]. For molecular visualization and crystallographic analysis, the *OLEX2*, version 1.3; OlexSys Ltd., Durham, UK, 2020 [17] was used. All non-H atoms were refined in the anisotropic approximation. Hydrogen atoms were placed onto calculated positions and refined isotropically within the riding model. The crystal structure contained an appreciable fraction (29%) of disordered hydrate water molecules that cannot be localized and refined using difference Fourier maps. They were taken into account in the refinement procedure using the Solvate Mask function of *OLEX2* [18].

Thus main crystallographic parameters for DOTA-Phe-D-Trp-Lys-Thr-OMe ($M$ = 980.10 g/mol): tetragonal, space group $P4_32_12$ (no. 96), $a$ = 20.335(3) Å, $c$ = 27.105(5) Å, $V$ = 11208(4) Å$^3$, $Z$ = 8, $T$ = 100(2) K, μ(0.80246 Å) = 0.12 mm$^{-1}$, $Dcalc$ = 1.344 g/cm$^3$, 163,239 reflections measured (3.198° $\leq 2\Theta \leq$ 61.826°), 12,317 unique ($R_{int}$ = 0.1580, $R_{sigma}$ = 0.0542) which were used in all calculations. Number of restraints—53, number of parameters—710, $\Delta\rho_{max}$, $\Delta\rho_{min}$ (e Å$^{-3}$)—0.47, −0.41. The final $R_1$ was 0.096 ($I$ > 2σ($I$)) and $wR_2$ was 0.1986 (all data). CCDC deposition number—2121101.

## 3. Results and Discussion

### 3.1. Structure Solution and Refinement

The preliminary choice of space group for the experimental diffraction set was not obvious. According to the systematic absences statistics shown in Table 1, we first tried to solve the structure in the $P4_2$ space group using *SHELXT* [15] in the mode of extended search for an appropriate space group within the given Laue class. A reasonable solution was obtained in the $P4_3$ space group with two symmetrically independent peptide molecules. At later stages of the structure refinement, we have run the ADDSYMM search as implemented in the *PLATON*, version 100419; University Of Glasgow, Glasgow, UK [19] software and converted the dataset to the $P4_32_12$ space group with only one symmetry independent molecule. No obvious signature of twinning was identified.

**Table 1.** Systematic absences statistics.

|  | $4_1/4_3$ | $4_2$ | *n-* | *-b-* | *-c-* | *-n-* | *-2$_1$-* | *-c* |
|---|---|---|---|---|---|---|---|---|
| $N$ | 93 | 59 | 1754 | 5254 | 5162 | 5192 | 87 | 3158 |
| $N \cdot I > 3$ s | 28 | 7 | 1005 | 2469 | 2347 | 2276 | 25 | 1604 |
| $<I>$ | 81.9 | 30.0 | 247.2 | 224.6 | 216.7 | 181.4 | 70.8 | 218.4 |
| $<I/\sigma>$ | 2.5 | 1.3 | 4.7 | 3.9 | 3.9 | 3.6 | 2.4 | 4.1 |

The refinement of the molecular geometry of the peptide fragment was relatively stable but best-fit values of $R_1/wR_2$ remained rather high about 13/27%. This observation was assigned to the presence of an appreciable fraction of poorly ordered hydrate water molecules. The Solvent Mask procedure of OLEX2 [18] identified the total volume of solvent accessible voids of 3269 Å$^3$ and 520 e$^-$ per unit cell, which amounts approximately 29% of the unit cell volume. Before using the Solvent Mask, two water molecules were isolated. Their positions were unambiguously determined from difference electron density maps. The occupancy of oxygen positions was fixed and equal to 1; this did not negatively affect the quality parameters of the model. In addition, hydrogen bonds between the

tetrapeptide and water were clearly observed. The final estimate for hydrate water content was approximately 8.5 molecules per tetrapeptide molecule. Alternatively, we tried to reveal water molecules in difference Fourier maps during refinement and finished up with a model containing nine $H_2O$ molecules with occupancies ranging from 1 to 0.38 summing up to 6.64 molecules per asymmetry part of the unit cell. The difference in 1.86 water molecules between the two models of the tetrapeptide structure may be because we could not determine all the positions of water in the second model, due to their strong disorder. Nevertheless, best results in terms of $R_1/wR_2$ were reached using the Solvate Mask procedure.

All hydrogen atoms were refined using a riding model with appropriate instructions AFIX 23 for -$CH_2$ group C-H = 0.99 Å, AFIX 43 for N-H = 0.88 Å and C-H = 0.95 Å (aromatic), AFIX 13 for C-H = 1.00 Å and N-H = 1.00 Å (DOTA), and a rotating model—FIX 137 -$CH_3$ group C-H = 0.98 Å, AFIX 147 for -OH = 0.84 Å. The $U_{iso}(H)$ = 1.5$U_{eq}$(parent atom) for the methyl and −OH groups, and 1.2$U_{eq}$(parent atom) otherwise. For the water molecules the instruction AFIX 6 was used, −OH = 0.87 Å. The distances $C_{45B}$-$C_{44}$, $C_{45B}$-$O_{11B}$, $C_{45B}$-$O_{10B}$, $O_{11B}$-$O_{10B}$ of only one carboxyl group of DOTA and $C_{10}$-$C_{11B}$, $C_{11B}$-$N_{2B}$, $C_{10}$-$H_{10A}$ of Lys were restrained by DFIX instruction: C-C and C-O = 1.5 Å, O-O = 2.3 Å, C-H = 1.0 Å with σ = 0.02 Å, C-N = 2.5 Å with σ = 0.04 Å. In addition, FLAT, SIMU, DELU and EADP instructions were used. Please see refinement description in .cif file from our Supplementary Materials, which can be downloaded use 2121101 CCDC number.

### 3.2. Structure Description and Crystal Structure Analysis

After determining the structure of the tetrapeptide DOTA-Phe-D-Trp-Lys(ε-Boc)-Thr-OMe, the similar structures containing the sequence——-Phe-Trp-Lys-Thr- were searched in a *Cambridge Structural Database (CSD)*, February 2021 version; Cambridge Crystallographic Data Centre: Cambridge, UK, 2021 [20] and in a RCSB PDB (https://www.rcsb.org/, 16 December 2021) [21] databases. The search for structures in *CSD* was done using the Crystal packing feature of *Mercury*, 2021.1 version; Cambridge Crystallographic Data Centre: Cambridge, UK, 2021 [22]. Thus, the structures of two octreotides were found [9,10] (Table 2).

**Table 2.** The structures of other somatostatin analogues containing the sequence -Phe-*D*-Trp-Lys-Thr.

| CCDC Ref. Code | RMS (Main Chain), Å |
|:---:|:---:|
| GOGZOU [9] | 1.50 |
| YICMUS [10] | 1.46 |
| **PDB Ref. Code** | **RMS (Main Chain), Å** |
| 1SOC [23] | 1.21 |
| 2SOC [23] | 0.896 |
| 2MI1 [24] | 0.946 |
| 6VC1 [25] | 1.23 |

The search in the PDB database was performed using the keywords "somatostatin analog" with further sorting of the results in the presence of the Phe-Trp-Lys-Thr sequence in the sequence of the found peptides. Thus, four structures [23–25] were found (Table 2), the sequences of the remaining somatostatin analogues did not contain the Phe-Trp-Lys-Thr sequence. One of these structures was determined by XRPD, three others by NMR.

The found structures were compared with the tetrapeptide using Structure Overlay of *Mercury* by atomic alignment of the peptides' main chains. The comparison results are shown in the Table 2.

In addition, 206 structures with the DOTA group were found in CCDC; RMS varied from 0.195 to 2.053 Å.

The molecular structure of DOTA-Phe-D-Trp-Lys($\varepsilon$-Boc)-Thr-OMe is shown in Figure 1. The side chain of the Lys residue and carboxylate groups of DOTA are disordered over at least two orientational positions. Occupancy of disordered parts are 0.740/0.260, 0.609/0.391, 0.587/0.413 for carboxylate groups of DOTA and 0.708/0.292 for amino group of Lys, respectively (Figure 1b). The occupancy of all disordered groups was determined using a free variable; moreover, for each group, a free variable was its own. Probably, this is the main reason for the intrinsic disorder of the network of hydrate water molecules. The asymmetric carbon atom C13 corresponds to the unnatural D-configuration of the Trp residue, which was deliberately introduced at the stage of synthesis to increase the proteolytic stability of the DOTA-Phe-D-Trp-Lys-Thr-OMe epitope.

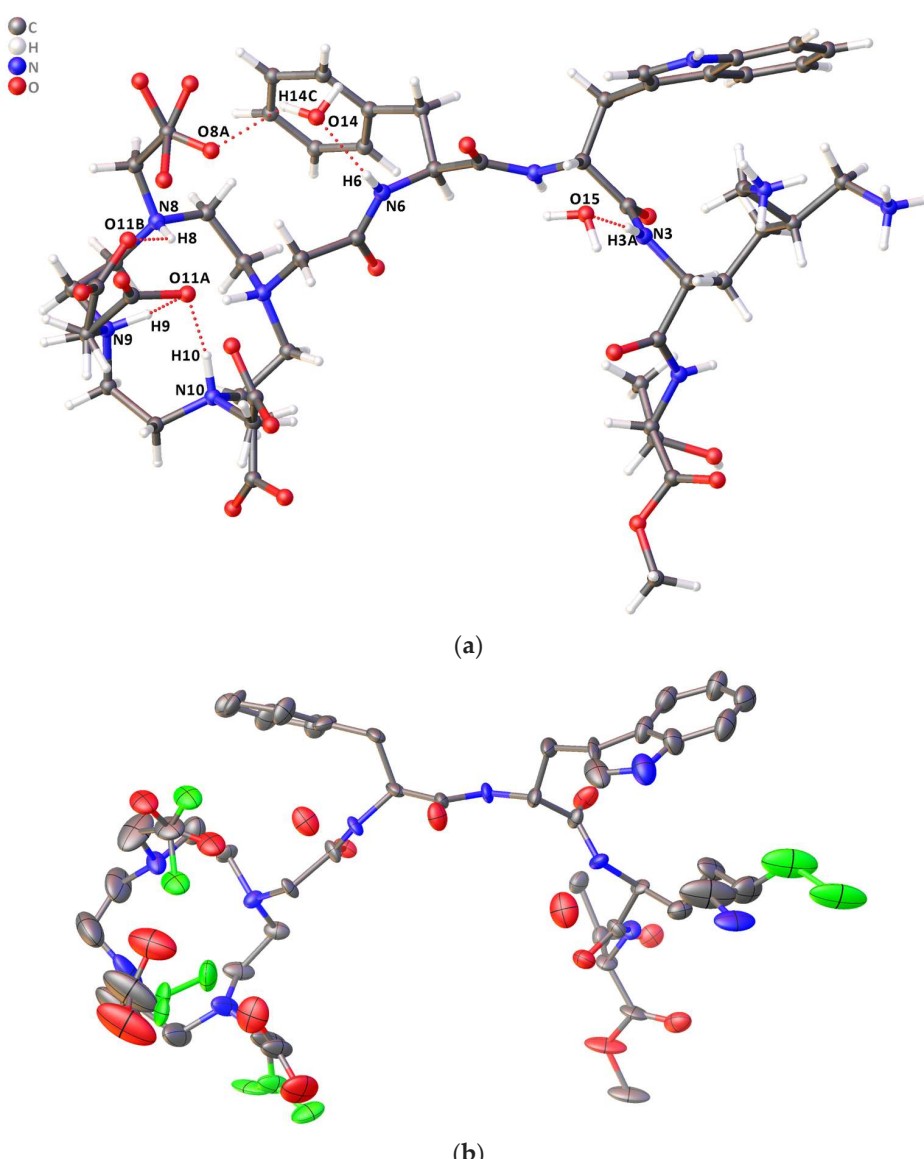

(a)

(b)

**Figure 1.** The molecular structure of DOTA-Phe-D-Trp-Lys-Thr-OMe: (**a**) three intramolecular H-bonds are observed in the tetapeptide molecule between DOTA atoms, but the presence of these H-bonds does not significantly affect the DOTA conformation. Protons at amino groups of the DOTA moiety are half occupied; (**b**) the molecular structure with ellipsoids, 50% level of probability. Two disordered orientations of carboxylate groups in the DOTA moiety and side chain with the terminal amino group of Lys are shown. The parts with less occupancy are colored green. Protons have been removed for clarity.

Three intramolecular H-bonds are observed in the DOTA fragment of a conjugate atoms: $N_9$-$H_9$...$O_{11A}$ and $N_{10}$-$H_{10}$...$O_{11A}$, besides the occupancy of oxygen $O_{11A}$ is 0.391, and $N_8$-$H_8$...$O_{11B}$, the occupancy of $O_{11B}$ is 0.609 and of protons $H_8$, $H_9$, $H_{10}$ was fixed at 0.5 (Figure 1a). The presence of these H-bonds does not significantly affect the DOTA conformation. Four symmetrically located intermolecular hydrogen bonds are more important in the structure of the tetrapeptide: between the amino and carboxyl groups of the D-Trp residue $N_5$-$H_5$...$O_5$ and between the amino group of the Thr residue and the carboxyl group of DOTA $N_1$-$H_1$...$O_7$. These H-bonds form the dimer (Figure 2). Two dimers bind to each other through H-bonds between two disordered groups: the amino group of Lys and the carboxyl group of DOTA ($N_{2A}$-$H_{2AC}$...$O_{10A}$); the indole nitrogen of D-Trp and the carboxyl group of DOTA ($N_4$-$H_4$...$O_{10B}$). These H-bonds form the crystal package shown in the Figure 3. In addition, one strong hydrogen bond $O_3$-$H_3$...$O_{9A}$ can be distinguished, which connects two other dimers. It was mentioned before that the positions of oxygens of two water molecules are clearly determined from difference electron density maps, but the positions of hydrogen atoms can be assumed only from supposed H-bonds with the tetrapeptide molecule. One water molecule ($H_{14C}$-$O_{14}$-$H_{14D}$) has three H-bonds with the tetrapeptide: $O_{14}$-$H_{14C}$...$O_{8A}$ (DOTA), $O_{14}$-$H_{14D}$...$O_4$ (Lys), and the other water molecule ($H_{15A}$-$O_{15}$-$H_{15B}$) has two hydrogen bonds: $N_{2B}$-$H_{2BA}$...$O_{15}$ (Lys), $N_3$-$H_{3A}$...$O_{15}$ (Lys). All H-bonds present in the structure are shown in Table 3.

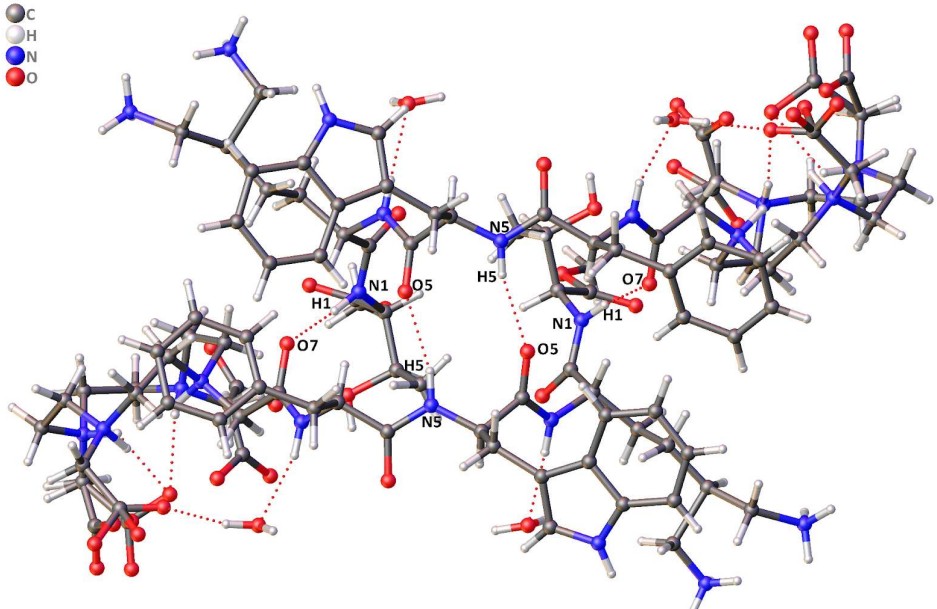

**Figure 2.** Intermolecular contacts in the crystal structure of DOTA-Phe-D-Trp-Lys-Thr-OMe: the N-H . . . O H-bond between. These H-bonds form a dimer.

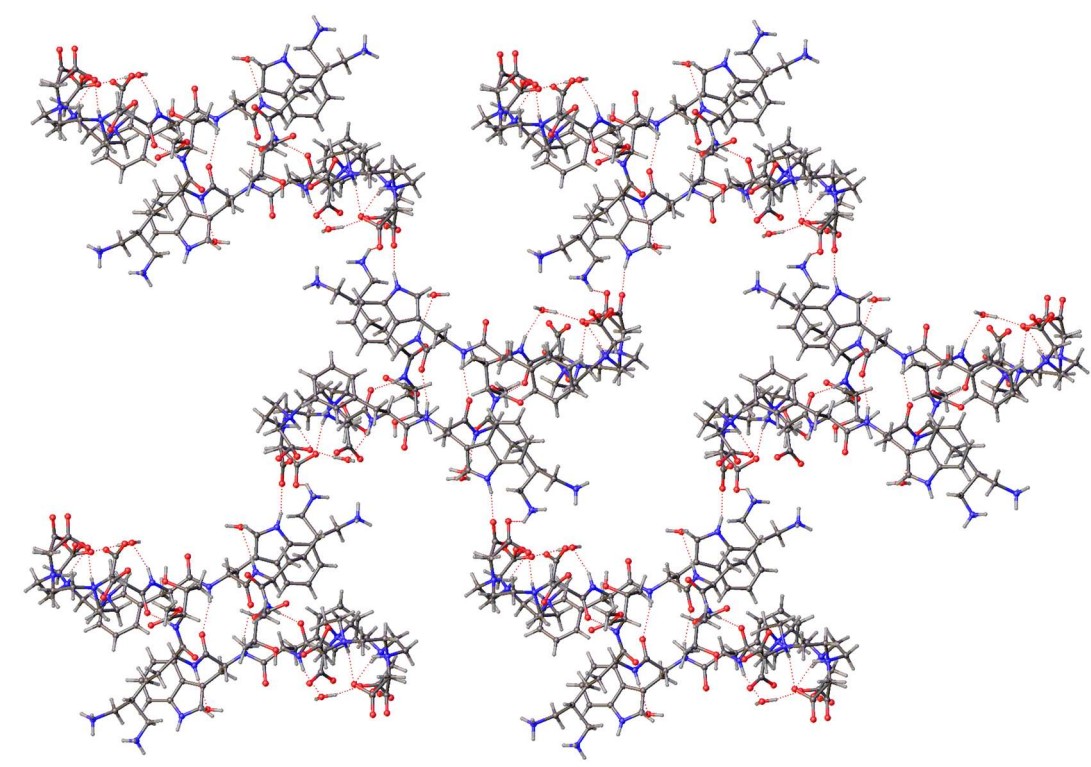

**Figure 3.** Cooperative intermolecular contacts in the crystal structure of DOTA-Phe-D-Trp-Lys-Thr-OMe. Two dimers bind to each other through H-bonds between two disordered groups: the amino group of Lys and the carboxyl group of DOTA; the indole nitrogen of D-Trp and the carboxyl group of DOTA. Crystalline packaging is observed.

**Table 3.** H-bonds are present in the structure of DOTA-Phe-D-Trp-Lys-Thr-OMe. Symmetry code(s): (i) −y + 3/2, x + 1/2, z − 1/4; (ii) −y + 3/2, x + 1/2, z + 3/4; (iii) y − 1/2, −x + 3/2, z + 1/4; (iv) −x + 3/2, y − 1/2, −z + 3/4; (v) y, x, 1 − z; (vi) 1 − x, 2 − y, z + 1/2.

| D–H...A | D–H | H–A | D–A | DHA |
|---|---|---|---|---|
| $O_3–H_3...O_{9A}{}^{iv}$ | 0.84 | 1.90 | 2.738 (8) | 179.8 |
| $O_3–H_3...O_{9B}{}^{iv}$ | 0.84 | 2.12 | 2.87 (2) | 147.3 |
| $N_1–H_1...O_7{}^{v}$ | 0.88 | 1.95 | 2.812 (7) | 165.0 |
| $N_3–H_{3A}...O_{15}$ | 0.88 | 2.04 | 2.922 (8) | 175.6 |
| $N_4–H_4...O_{10B}{}^{vi}$ | 0.88 | 1.95 | 2.791 (15) | 159.5 |
| $N_5–H_5...O_5{}^{v}$ | 0.88 | 2.16 | 3.017 (7) | 163.7 |
| $N_6–H_6...O_{14}$ | 0.88 | 2.09 | 2.934 (6) | 159.3 |
| $N_8–H_8...O_{11B}$ | 1.00 | 2.26 | 3.003 (14) | 129.8 |
| $N_9–H_9...O_{11A}$ | 1.00 | 2.08 | 2.739 (15) | 121.4 |
| $N_{10}–H_{10}...O_{11A}$ | 1.00 | 2.14 | 3.036 (18) | 148.2 |
| $N_{2A}–H_{2AC}...O_{10A}{}^{vi}$ | 0.91 | 2.14 | 2.728 (17) | 121.9 |
| $N_{2B}–H_{2BA}...O_{15}{}^{iii}$ | 0.91 | 1.98 | 2.72 (3) | 136.7 |
| $N_{2B}–H_{2BC}...O_{11B}{}^{ii}$ | 0.91 | 2.13 | 3.01 (4) | 161.0 |
| $O_{14}–H_{14C}...O_{8A}$ | 0.85 | 2.02 | 2.667 (9) | 132.1 |
| $O_{14}–H_{14D}...O_4{}^{i}$ | 0.85 | 1.91 | 2.731 (7) | 161.8 |
| $O_{15}–H_{15A}...O_{14}{}^{iii}$ | 0.87 | 2.02 | 2.731 (8) | 138.1 |

### 3.3. Computational Analysis of the Peptide Backbone Conformation

Notably, in the solved crystal structure, the peptide chain appears in an "open" conformation, which is held by numerous intermolecular interactions. At the same time, in the NMR-resolved solution-state structure of octreotide it adapts a β-hairpin structure [24]. To determine the peptide chain conformation of the synthesized DOTA-Phe-D-Trp-Lys-Thr-OMe in an application-ready complex with a lanthanide ion in solution, we have performed conformational search for the corresponding associate with $Lu^{3+}$ using MacroModel [26,27] (force field: OPLS3, permittivity: 81, optimization method: optimal). The conformers within 6 kcal/mol from the lowest-energy one (15 items) were optimized in Gaussian16 [28] at PBE0 [29]-D3BJ [30,31]/def2SVP [32,33]/PCM [34] ($H_2O$) level of theory. Optimized structures with corresponding energies are provided as SI in Structures.xyz.

Figure 4 shows molecular structure of the lowest-energy conformation of $Lu^{3+}$-DOTA-Phe-D-Trp-Lys-Thr-OMe complex according to quantum chemical calculations (left), as well as its overlay (red) with NMR-resolved structures of octreotide [25] (red) and somatostatin [24] (blue). Notably, all three compounds have very similar backbone structures, with remarkable resemblance between tetrapeptide chains of the computed compound and octreotide, having RMSD of only 0.2 Å between the four alpha-carbons.

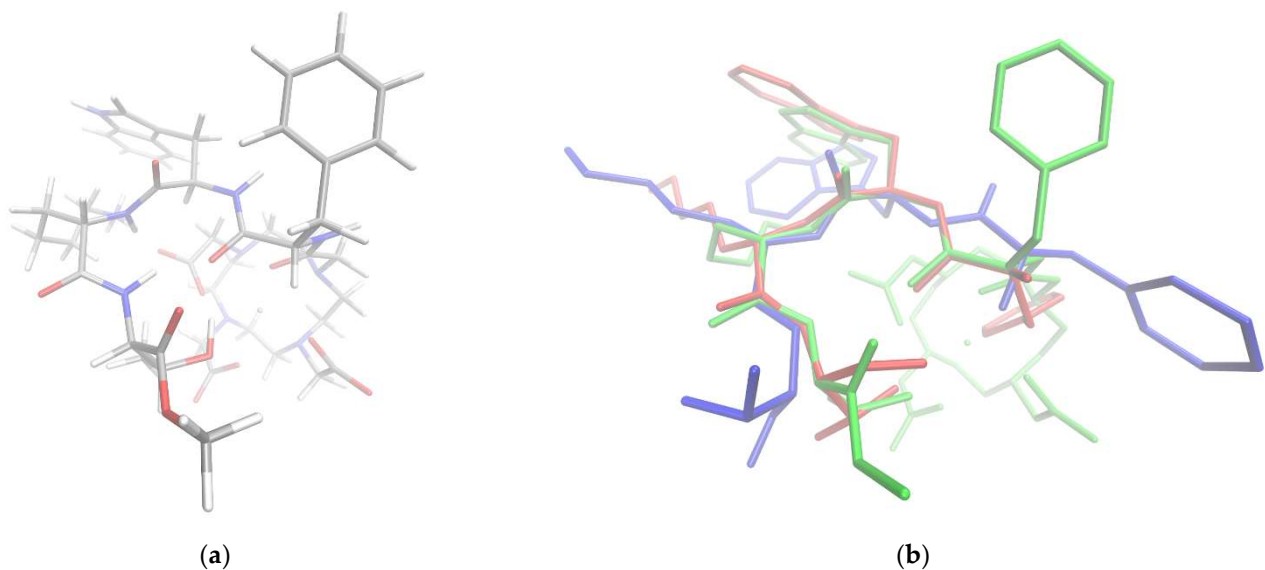

(**a**)　　　　　　　　　　　　　　　　　　　　　　　(**b**)

**Figure 4.** (**a**) Lowest-energy conformation of $Lu^{3+}$-DOTA-Phe-D-Trp-Lys-Thr-OMe according to PBE0-D3BJ/def2SVP/PCM($H_2O$) level of theory; coloring: gray—carbon, blue—nitrogen, red—oxygen, white—hydrogen, pink—lutetium. (**b**) The same conformation (green) overlaid with NMR-resolved structures of octreotide (ref. code of PDB 1SOC) (red) and somatostatin (ref. code of PDB 2MI1) (blue); hydrogen atoms are omitted for clarity.

Thus, although DOTA-Phe-D-Trp-Lys-Thr-OMe adopts an open conformation in crystal, its complex with $Lu^{3+}$ in water prefers a β-hairpin structure identical to that in octreotide. This result signifies that, unlike for protein structures which are held together by tens-to-hundreds hydrogen bonds and other noncovalent interactions, X-ray diffraction studies cannot be regarded as a reliable source of native (i.e., biologically relevant) structures of small peptides. This makes molecular modeling an essential tool for studying the native conformations of crystallized peptides.

### 4. Conclusions

A conjugate of a short somatostatin analog of high purity has been synthesized and successfully crystallized. The crystal structure of the conjugate was determined by X-ray diffraction analysis, which is the first structural report for a vector molecule for a radiopharmaceutical and for short peptide somatostatin analogs. The crystal structure and

all H-bonds were fully described in this work. In addition, the structure of tetrapeptide was compared with other somatostatin analogs containing the sequence Phe-*D*-Trp-Lys-Thr. Only three structures of the found somatostatin analogs were determined by X-ray diffraction method.

While in the solved crystal peptide chain appears in an unnatural "open" conformation interacting with the surrounding molecules, quantum chemical modeling has shown that upon solvation and lanthanide cation ($Lu^{3+}$) binding it adopts the β-hairpin structure, identical to that of octreotide and other cyclic peptide analogs of somatostatin. This result is one of the fundamental factors for the use of the synthesized epitope in nuclear medicine. It also signifies that, unlike for protein structures which are held together by tens to hundreds of hydrogen bonds and other noncovalent interactions, X-ray diffraction studies cannot be regarded as a reliable source of native (i.e., biologically relevant) structures of small peptides. This makes molecular modeling an essential tool for studying the native conformations of crystallized peptides.

**Supplementary Materials:** The following are available online at https://www.mdpi.com/article/10.3390/cryst12010012/s1, FWkT.xyz file contain optimized atomic coordinates of molecular models predicted by quantum chemical simulation as energy minima together with respective absolute energies; crystallofrafic information file (.cif) of DOTA-Phe-D-Trp-Lys-Thr-OMe can be downloaded from CSD use 2121101 CCDC number.

**Author Contributions:** Conceptualization, Y.Z. and D.A.; data curation, P.D.; methodology A.N., M.O and M.S.; valida-tion, S.A. and S.D.; investigation, S.D., S.A., V.L., D.K. and Y.Z.; resources, D.A., P.D., V.L., D.K., M.M. and A.N.; software, D.A. and M.M.; writing—original draft preparation, S.D.; writing—review and editing, Y.Z.; visualization, S.A., S.D. and M.M.; project administration, Y.Z.; funding acqui-sition, S.A. and Y.Z. All authors have read and agreed to the published version of the manuscript.

**Funding:** This research was partially funded by Russian Ministry of Science and Education via the budget project of SRF SKIF, Boreskov Institute of Catalysis and by the Ministry of Science and Higher Education of the Russian Federation in the implementation of the research program "Use of synchrotron radiation for virologic research" within the framework of the Federal Scientific and Technical Program for the Development of Synchrotron and Neutron Research and Research Infrastructure for 2019–2027 (Agreement No. 075-15-2021-1355 (12 October 2021)).

**Institutional Review Board Statement:** Not applicable.

**Informed Consent Statement:** Not applicable.

**Data Availability Statement:** The datasets generated and analysed during the current study, including raw diffraction images, are available from the corresponding author on reasonable request.

**Acknowledgments:** This work has been carried out using computing resources of the federal collective usage center Complex for Simulation and Data Processing for Mega-science Facilities at NRC "Kurchatov Institute", http://ckp.nrcki.ru/, 16 December 2021. Mass spectrometric studies were carried out on the equipment of the NRC "Kurchatov Institute"—The main collective research bases of IREA.

**Conflicts of Interest:** The authors declare no conflict of interest.

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
