# Peer review of "The Crystal Structure Elucidation of a Tetrapeptide Analog of Somatostatin DOTA-Phe-D-Trp-Lys-Thr-OMe"

_crystals, doi:10.3390/cryst12010012_

Round 1
Reviewer 1 Report
In this manuscript, Diusenova et al solved the crystal structure of a tetrapeptide analogue of somatostatin using x-ray diffraction methods. They described intramolecular interactions and interactions with H2O solvent of this peptide and noted that it adopts an open conformation that is different from similar somatostatin analogues, which exhibits beta hairpin confirmation. They then modeled their peptide analogue with presence of Lu3+ and pointed out that it could potentially prefer beta hairpin structure as well.
There are some questions and concerns for this manuscript:
- Figure 3 in this manuscript is missing entirely;
- Since Lu3+ is important for the peptide analogue to adopt relevant conformation, it is puzzling why the authors did not crystallize the peptide with Lu3+ present in the first place. The open confirmation of the peptide might due to crystal packing, thus might not be biologically relevant;
- Overall, the structure of a peptide epitope without the context of its cognate interactions reveals minimal information on how it actually works, so the significance of this manuscript is questionable. For the above reasons I do not recommend the publication of this manuscript.
Author Response
Dear reviewer, please see our answers in the attachment.

Reviewer 2 Report
The paper presents the results of the crystallographic and molecular dynamic study of the N- and O-terminal protected tetrapeptide considered to represent an epitope for hormone somatostatin.
The Authors find that the fragments of the structure are disordered, which has a consequence in the presence of a significant fraction of poorly ordered solvent (water molecules) network in the crystal lattice. This has an effect in a relatively high R1-factor in final refinement.
The paper is very well written, and the findings clearly described. I recommend to accept the paper after minor corrections. My comments are provided below.
1. page 2, lines 86-87: 'shock-freezed' could be said as 'flash-frozen'
2. pg2 ln91: 'X-ray radiation with (λ = 0.80246 Å was used:'
parenthesis are not needed before lambda, should be: 'X-ray radiation with λ = 0.80246 Å was used'
3. pg3 ln 111: Δρmax, Δρmin: 'max' and 'min' could be gigen in subscript.
4. pg5 ln172, and pg6 ln197: Figures 1 and 2: The two disorder components are not easy to distinguish in the Figures. Is it possible to show them, for example, with use of different color scheme.
5. Figure 1 is not mentioned in the text.
6. I can not find Figure 3, alhough it is mentioned in the text. Figures 1 and 2 are followed by Figure 4.
7. pg8 ln 253: The entire paragraph 'Conflicts of Interest" should be checked to provide appropriate information.
8. pg9 ln 308: In Reference number 20, the doi number (doi: 10.1093/nar/28.1.235) is missing.
Author Response
Dear reviewer, thanks for your comments. Please find our answers in the attachment.

Reviewer 3 Report
Diusenova et al. report the single crystal X-ray structural analysis of a tetrapeptide as an analogue of Somatostatin. Details of the molecular structure and intermolecular interactions are discussed and are further complimented by a computational analysis.
Single crystal molecular structures of oligo or polypeptides, which are on the verge between small molecules and macromolecules, and in particular those of non-cyclised linear peptides are comparatively rare due to their inherent flexibility. The presented work is therefore quite an achievement and certainly worthy of being published in Crystals.
A few issues should be considered, though, during a minor revision:
- Decide whether it should be “analog” or “analogue” and use it uniformly throughout.
- Line 45: “… and drugs based on it …”, please specify whether you are really referring to derivatives of somatostatin or whether you are referring to drugs inspired by the structure of somatostatin or even both.
- Line 51/52: “…and selectively acting on human 51 lung adenocarcinoma … “. This sentence is grammatically wrong. It is probably meant “ … which selectively act on …”
- Line 62/63: “This article represents the very first structural report on a somatostatin analog molecule.” This contradicts reference 10. Please correct/specify.
- 1. Synthesis: Where does the DOTA come from? Supplier? Synthesised?
- Line 111/112: The R values given here appear to be higher (worse) than in the cif file. Please check carefully (also all other data in this section).
- Provide at least one figure of the molecular structure with ellipsoids (50% or 30% level of probability).
- Line 152: replace “didn’t” by “did not”
- Caption of Figure 1: Two full stops at the end.
- Line 187: clarifly (????), clearly (?)
- Line 198: use the plural form: H-bonds
- Lines 204/209/216: The “3+” should be superscript
- Line 225: replace “proved” by “determined”
- Line 227: The hairpin statement refers only to the computational results. This should be stated more clearly. In fact, this sentence might be better placed in the second paragraph of the conclusion.
Crystallographic issues:
- The structure must be moved into the unit cell. It is now entirely outside.
- An explanation how the water hydrogen atoms were treated is required. There is not sufficient electron density around O15 to suggest that both H atoms can be found on the map. In general hydrogen atoms on water should not be added without crystallographic evidence for their presence. In particular with such a large number of H-bonding, acidic and basic sites this is critical. If not all respective hydrogen atoms can be located, this emphasizes H mobility within the structure, which is not unexpected and that should be simply described as is; i.e. factual. It is similarly problematic that all N-bound nitrogen atoms are fixed in particular for the DOTA moiety. Was this protonation state supported by electron density? A more comprehensive description of the H treatment should be provided in the experimental part.
- For the solvent mask the fundamental work, with which such strategy was introduced, should be cited: Van Der Sluis, A. L. Spek, Volume 46, Issue3, March 1990, Pages 194-201, https://doi.org/10.1107/S0108767389011189
- A more comprehensive description of how the disorders were modelled should be provided.
Author Response
Dear editor,
please find our answers in the attachment.
